# Visionary-R1: Mitigating Shortcuts in Visual Reasoning with Reinforcement Learning

## Abstract

Learning general-purpose reasoning capabilities has long been a challenging problem in AI. Recent research in LLMs, such as DeepSeek-R1, has shown that reinforcement learning techniques like GRPO enable pre-trained LLMs to develop reasoning capabilities using simple question-answer pairs. In this paper, we aim to train visual language models (VLMs) to perform reasoning on image data through reinforcement learning and visual question-answer pairs, without explicitly using any chain-of-thought (CoT) supervision. Our key finding indicates that simply applying GRPO to a VLM—by prompting the model to think step by step before giving an answer—may cause the model to develop shortcuts from easy questions, resulting in poor generalization of reasoning to broader question domains. We argue that the key to mitigating shortcut learning is to encourage the model to interpret images prior to reasoning. Therefore, we train the model to adhere to a caption-reason-answer output format: initially generating a detailed caption for an image, followed by constructing an extensive reasoning chain. When trained on 273K CoT-free visual question-answer pairs and using only reinforcement learning, our model, named Visionary-R1, outperforms strong multimodal models (e.g., GPT-4o, Claude3.5-Sonnet, and Gemini-1.5-Pro) on multiple visual reasoning benchmarks. Code and models will be open-sourced.

## 1 Introduction

Reasoning is essential for enabling AI to tackle complex problems and make informed decisions in real-world applications. However, training AI models to reason is extremely challenging—primarily due to the lack of large-scale human-annotated reasoning data (Lightman et al., 2023; Christiano et al., 2017; Ouyang et al., 2022). Recent advances in large language models (LLMs), such as DeepSeek-R1 (Guo et al., 2025a), have demonstrated the potential to induce reasoning capabilities in LLMs via reinforcement learning and using only question-answer pairs, without explicit step-by-step supervision. Meanwhile, the computer vision community has begun exploring RL approaches for visual language models (VLMs), using methods like GRPO (Shao et al., 2024) to extend reasoning to multimodal settings (Meng et al., 2025; Feng et al., 2025; Liu et al., 2025; Shen et al., 2025). While these efforts are promising, existing visual reasoning models often rely on complex multi-stage training pipelines that are both computationally expensive and time-consuming. Moreover, these models heavily rely on labeled chain-of-thought (CoT) reasoning data distilled from proprietary models like GPT-4o, which is costly.

In this paper, we aim to lower the development cost of training VLMs for visual reasoning by using only reinforcement learning and paired visual question-answer data, *without relying on any CoT supervision*. Inspired by DeepSeek-R1, we adapt GRPO to training VLMs using only question-answer pairs. Specifically, given an image and a question, we prompt a VLM to generate a reasoning chain followed by an answer and optimize the model using a combination of an accuracy reward (that evaluates the answer correctness) and a format reward (that encourages the reason-answer output format). However, this seemingly straightforward setup leads to a critical failure mode: the model develops *shortcuts* by producing short, uninformative reasoning chains. These shortcuts often suffice to answer easy training questions correctly, but the model fails to generalize to harder questions that require genuine visual understanding.

**Easy Sample (Train)**

Q: How many hospitals in Colorado were owned by the state or local government in 2019? A: 26

**GRPO**

<think> The number of hospitals in Colorado owned by the state or local government in 2019 is 26.</think> <answer> 26 </answer>

**Visionary-R1**

<info> The image is a bar graph that displays the number of hospitals in Colorado by ownership type. Here are the key points:
- The total number of hospitals in Colorado is 90.
- 26 hospitals are owned by the state/local government.
- 46 hospitals are non-profit.
- 18 hospitals are for-profit.
</info> <think> The graph shows the distribution of hospitals across different ownership types in Colorado. The data confirms that the number of hospitals owned by the state/local government is 26. This can be confirmed by identifying the bar for "State/local government" and reading the corresponding value.
</think> <answer> 26 </answer>

**Hard Sample (Test)**    Q:How many queries have a p-value lower than 0.50? A:4

|         | shutdown | bullet | objective | superintelligent | rot23 | paperclips | paperclips_large | water |
|---------|----------|--------|-----------|------------------|-------|------------|------------------|-------|
| p-value | **0.92** | **0.69** | 0.01    | **0.10**         | **0.58** | **0.43** | **0.60**        | 0.01  |

Table 1: **Goodness-of-fit results for power law models.** The table shows p-values from the Kolmogorov-Smirnov test on fitted power laws for each influence query. Values above the 0.1 thresholds suggested by Clauset et al. (2009) indicate the power law is a plausible fit.

**GRPO**

<think> To answer the question, we need to count the number of p-values that are lower than 0.50. </think>
<answer> 3 </answer>

**Visionary-R1**

<info> The image contains a table titled "Table 1: Goodness-of-fit results for power law models." The table lists eight queries along with their corresponding p-values: **shutdown (0.92), bullet (0.69), objective (0.01), superintelligent (0.10), rot23 (0.58), paperclips (0.43), paperclips_large (0.60), and water (0.01).** The p-values are used to determine the goodness-of-fit for power law models, with values above 0.1 suggested by Clauset et al. (2009) indicating a plausible fit. </info> <think> To determine how many queries have a p-value lower than 0.50, we need to count the number of p-values that are less than 0.50. From the table, the p-values are: 0.92, 0.69, 0.01, 0.10, 0.58, 0.43, 0.60, and 0.01. The p-values that are less than 0.50 are 0.01, 0.01, 0.1, and 0.43. Therefore, there are four queries with p-values lower than 0.50. </think> <answer> 4 </answer>

Figure 1: Comparison between the GRPO model and Visionary-R1. Using the reason-answer output format, the GRPO model tends to generate shortcut responses for easy samples during training, which hinders the model from learning general-purpose reasoning capabilities and results in poor generalization performance. In contrast, with a more comprehensive understanding of the image context, i.e., using the caption-reason-answer output format, Visionary-R1 consistently generates long, meaningful reasoning for both easy and hard samples.

As illustrated in Fig. 1, the model trained with GRPO performs well on simple training examples with shortcuts (top), but at test time, it produces incoherent reasoning and incorrect answers on unseen examples (bottom).

To address the shortcut issue, we propose **Visionary-R1**, a reinforcement learning framework that enforces visual understanding before reasoning. The key idea is to train the model in a structured caption–reason–answer format, where it must first generate a detailed caption of the image before reasoning and answering. The captioning step ensures that the model does not just rely on superficial cues or patterns but engages in a deeper analysis of the image context, regardless of whether the question is easy or hard—this forces the model to adopt a consistent problem-solving approach, thus mitigating potential shortcuts and consequently making the reasoning capabilities more generalizable across different data distributions. To en-

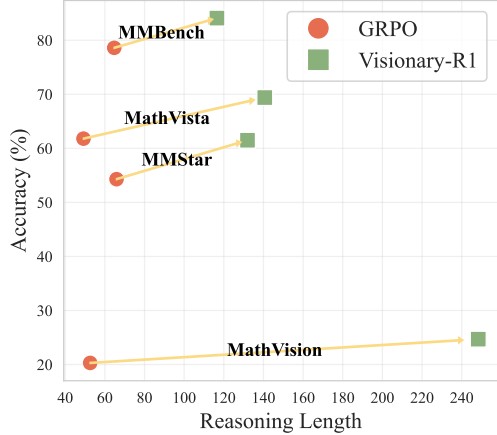

Figure 2: The longer the reasoning chain, the better the accuracy.

sure the caption is informative, we impose auxiliary supervision on the caption tokens by using reinforcement learning from AI feedback (Bai et al., 2022). This caption reward is combined with standard accuracy and format rewards during policy optimization. The resulting model produces longer, more meaningful reasoning tokens than the model learned with GRPO alone (Fig 1), leading to better generalization performance on unseen data (Fig 2).

Our contributions can be summarized as follows:

- We identify a critical issue in training VLMs to reason via RL, i.e., the model tends to develop shortcuts (short, uninformative reasoning), which significantly limit generalization ability.

- We propose a simple RL approach, which asks the model to adopt a caption-reason-answer output format and uses an AI-guided caption reward to force the model to interpret images before reasoning.

- We conduct extensive experiments on challenging visual reasoning benchmarks, including MathVista, MathVision, and MMBench. Despite using only 3B parameters and training on CoT-free question-answer pairs, our model outperforms strong commercial models like GPT-4o and Claude3.5-Sonnet.

## 2 Related Work

### 2.1 Vision-Language Models

Vision-language models (VLMs) have made substantial progress in multimodal perception, instruction following, and question answering. Proprietary systems such as GPT-4V (OpenAI, 2023), Claude (Anthropic, 2024), and Gemini (Team et al., 2024), as well as open-source models such as Qwen2.5-VL (Bai et al., 2025), InternVL2.5 (Chen et al., 2024b), and LLaMA3.2 (AI, 2024), have established strong performance across a wide range of benchmarks.

Despite these advances, most high-performing VLMs are optimized primarily through supervised fine-tuning, which does not necessarily induce robust multi-step reasoning. Recent reasoning-oriented VLMs often rely on chain-of-thought (CoT) distillation or curated reasoning traces (Yang et al., 2025; Dong et al., 2024; Zhang et al., 2025; Xu et al., 2024). Such pipelines can improve benchmark accuracy, but they depend on annotated or distilled intermediate supervision that is expensive to scale and difficult to reproduce. In contrast, we study whether visually grounded reasoning can be learned directly from question-answer supervision.

### 2.2 Reinforcement Learning

Reinforcement learning has become a powerful tool for improving reasoning in language models. DeepSeek-Math (Shao et al., 2024) introduced Group Relative Policy Optimization (GRPO), which removes the need for an explicit critic by normalizing rewards within a sampled group of responses. DeepSeek-R1 (Guo et al., 2025a) further demonstrated that answer-driven RL can elicit strong reasoning behavior from large language models without explicit step-by-step supervision.

These results motivate extending RL to multimodal models, but the visual setting introduces an additional challenge: the policy must not only predict the correct answer, but also ground its intermediate computation in image evidence. Answer-level optimization alone may therefore be insufficient once perception becomes part of the reasoning process. Our work focuses on this gap by designing reward signals that encourage grounded intermediate representations rather than answer-only optimization.

### 2.3 Visual Reasoning

Visual reasoning benchmarks such as MathVista (Lu et al., 2023), MathVision (Wang et al., 2024), and MMBench (Liu et al., 2024) evaluate a diverse set of capabilities, including mathematical reasoning, perception of fine-grained visual attributes, and multimodal knowledge integration. Prior work has improved performance through stronger multimodal backbones, more effective cross-modal integration, CoT distillation, or specialized reasoning supervision (Yang et al., 2025; Dong et al., 2024; Zhang et al., 2025; Xu et al.,

2024; Chen et al., 2024b; Yao et al., 2024b; Yan et al., 2025; Yao et al., 2024a). Representative examples include Vision-R1 Huang et al. (2025), which distills reasoning trajectories from DeepSeek-R1 to construct a 200K-sample cold-start dataset. VL-Rethinker Wang et al. (2026) explicitly inserts trigger tokens, such as "Wait," into the reasoning chain to encourage more extended deliberation.

However, existing approaches typically optimize either final-answer correctness or annotated reasoning quality, with limited attention to whether RL-trained trajectories remain visually grounded. The Look-Back Yang et al. (2026) method attempts to incorporate more visual information into the reasoning process, but still relies on external models such as GPT-4o to provide the CoT training data. We instead emphasize shortcut mitigation and show that explicit caption-level supervision can improve transfer under QA-only reinforcement learning, without the need for additional model-distilled data or extra supervision.

## 3 Method

### 3.1 Problem Formulation

Given a visual question-answering dataset $\mathcal{D} = \{(v_i, q_i, a_i)\}_{i=1}^{N}$, where $v_i$ is an image, $q_i$ a question, and $a_i$ the answer, the goal is to train a VLM policy $\pi_\theta$ to produce informative reasoning processes that lead to correct answers. Unlike prior work that relies on labeled reasoning data (often obtained through model distillation), we assume access only to question-answer pairs. We define $x = (v, q, a)$ as a sample and the observable model input as $u = (v, q)$. Conditioned on $u$, the policy samples a rollout $o = (r, y) \sim \pi_\theta(\cdot \mid u)$, where $r$ is the reasoning process and $y$ the predicted answer.

Because the supervision considers only the final answer, the most straightforward performance metric is expected answer accuracy:

$$\mathrm{Acc}(\pi_\theta) = \mathbb{E}_{x \sim \mathcal{D}} \, \mathbb{E}_{o \sim \pi_\theta(\cdot \mid u)} \big[ \mathrm{Eval}(y, a) \big], \tag{1}$$

where $\mathrm{Eval}(y, a) = 1$ if the predicted answer $y$ is correct and 0 otherwise. In the standard answer-driven RL setting, and suppressing the dependence on $x$ for notational simplicity, this correctness signal is directly used as the reward, i.e., $R(o) = \mathrm{Eval}(y, a)$.

Given sampled rollouts, RL updates the policy by optimizing a loss defined from the reward. In its generic form, the training loss can be written as

$$\mathcal{L}_{\mathrm{RL}}(\theta) = -\mathbb{E}_{x \sim \mathcal{D}, \, o \sim \pi_\theta(\cdot \mid u)} \big[ R(o) \big]. \tag{2}$$

and the parameters are updated by gradient-based optimization, i.e., $\theta \leftarrow \theta - \eta \, \nabla_\theta \mathcal{L}_{\mathrm{RL}}(\theta)$ where $\eta$ is the learning rate.

### 3.2 Motivation: Quantifying the Shortcut Phenomenon

GRPO (Shao et al., 2024) has proven effective for textual reasoning, but applying the same answer-driven optimization directly to visual reasoning introduces a critical failure mode. A VLM trained with GRPO can achieve competitive answer accuracy while its reasoning trace remains only weakly referred to the image, while relying on question-side regularities, answer-frequency biases, or superficial response patterns.

We refer to this behavior as a *shortcut*: the model reaches a plausible or even correct answer without genuinely conditioning on the visual input. As illustrated in Fig. 1, such behavior may be sufficient on relatively easy examples, but it does not constitute authentic image-grounded reasoning and tends to break on harder cases that require fine-grained visual evidence.

To make this phenomenon measurable, we explicitly connect *shortcut* to image–reasoning relevance. Intuitively, if a reasoning trace is well supported by visible evidence in the image, then the model is less likely to generate a shortcut; conversely, if the reasoning has weak relevance to the image, then the resulting prediction is more likely to be driven by textual cues in the question or other superficial patterns. We therefore define a *shortcut score* as the complement of an image–reasoning relevance score. Formally, let $x = (v, q, a)$ denote an image-question-answer triple, let $o = (r, y)$ denote a sampled rollout, where $r$ is the reasoning trace and $y$ is the predicted answer, and let $\mathcal{E} = \{(x_j, o_j)\}_{j=1}^{M}$ denote the set of analyzed rollout pairs. For

Table 1: Shortcut analysis on the A-OKVQA and ChartQA test sets. An external VLM evaluator (GPT-5) estimates image–reasoning relevance, from which shortcut scores are derived. We report GRPO, its correct/failure splits, a caption-based intervention on the original failures, and Visionary-R1. The caption is generated by GPT-5 only for this diagnostic intervention and is not used for training.

| Setting | | Metrics | | |
|---------|---------|-----------|------|-----------|
| Method | Evaluation split | Shortcut↓ | Acc↑ | MathVista↑ |
| GRPO | All test samples | 69.2 | 80.7 | 61.8 |
| | Correct subset | 65.1 | - | - |
| | Failure subset | 84.7 | - | - |
| GRPO+Caption | All test samples | - | 83.3 | 65.2 |
| | Re-run on original failures | - | 15.8 | 18.9 |
| Visionary-R1 | All test samples | **4.5** | **84.6** | **69.4** |

each pair $(x, o) \in \mathcal{E}$, we use an external VLM evaluator $g_\phi$ (GPT-5), prompted with the image, question, and generated reasoning under a fixed rubric, to return a normalized image–reasoning relevance score in $[0, 1]$:

$$s_{\text{vis}}(x, o) = g_\phi(v, q, r). \tag{3}$$

A score of 1 means that the key reasoning steps are strongly supported by visible evidence in the image, whereas a score of 0 means that the reasoning is largely unrelated to the image. Accordingly, a larger $s_{\text{vis}}(x, o)$ indicates that the reasoning depends more on image evidence and is therefore less likely to reflect shortcut behavior. We define the shortcut degree of one rollout as the complement of this relevance score, i.e., $1 - s_{\text{vis}}(x, o)$, and the dataset-level shortcut score as its average over the analyzed rollout pairs:

$$\mathcal{S}(\mathcal{E}) = \frac{1}{|\mathcal{E}|} \sum_{(x,o) \in \mathcal{E}} \left(1 - s_{\text{vis}}(x, o)\right). \tag{4}$$

To ensure that this measurement is not an artifact of a particular evaluator, we further repeat the shortcut analysis with multiple strong VLM evaluators. Across different evaluators, we observe the same qualitative patterns: GRPO failures consistently exhibit substantially higher shortcut scores than correct predictions, caption-based visual evidence consistently improves downstream performance on the original failure cases, and Visionary-R1 consistently yields much lower shortcut scores. These consistent trends indicate that the proposed shortcut evaluation is reliable and reproducible. We provide the per-evaluator results in Table 11, and the exact evaluation prompts in the supplementary material to facilitate reproducibility. This formulation enables three complementary diagnostics, each targeting a different question. 1) The overall shortcut score measures how prevalent shortcut behavior is on the test set. 2) The gap between correct and failure subsets tests whether shortcut is systematically associated with wrong predictions. 3) The caption-based intervention asks whether supplying missing visual evidence can recover downstream performance, providing stronger evidence that insufficient use of image evidence is not merely correlated with failure, but is an important cause of it. To make the comparison concrete, Table 1 reports the overall GRPO result, the correct and failure subsets under GRPO, a caption-based intervention that re-runs the original GRPO failures with a GPT-5-generated detailed caption, and the overall result of Visionary-R1.

Table 1 yields three observations. First, shortcut is substantially more pronounced in error cases: under GRPO, the average shortcut score increases from 65.1% on correct samples to 84.7% on failure samples, indicating that incorrect predictions are much more likely to be associated with image-agnostic reasoning. Second, making visual information explicit improves downstream performance: augmenting GRPO with a detailed visual caption raises answer accuracy from 80.7% to 83.3% and MathVista from 61.8% to 65.2%; when we re-run the original GRPO failure subset, the same intervention recovers 15.8% answer accuracy and an 18.9% MathVista score. These gains provide direct evidence that access to visual evidence matters for successful reasoning. Third, Visionary-R1 not only improves end-task performance, reaching 84.6% answer accuracy and 69.4% on MathVista, but also sharply reduces the overall shortcut score to 4.5%. Taken

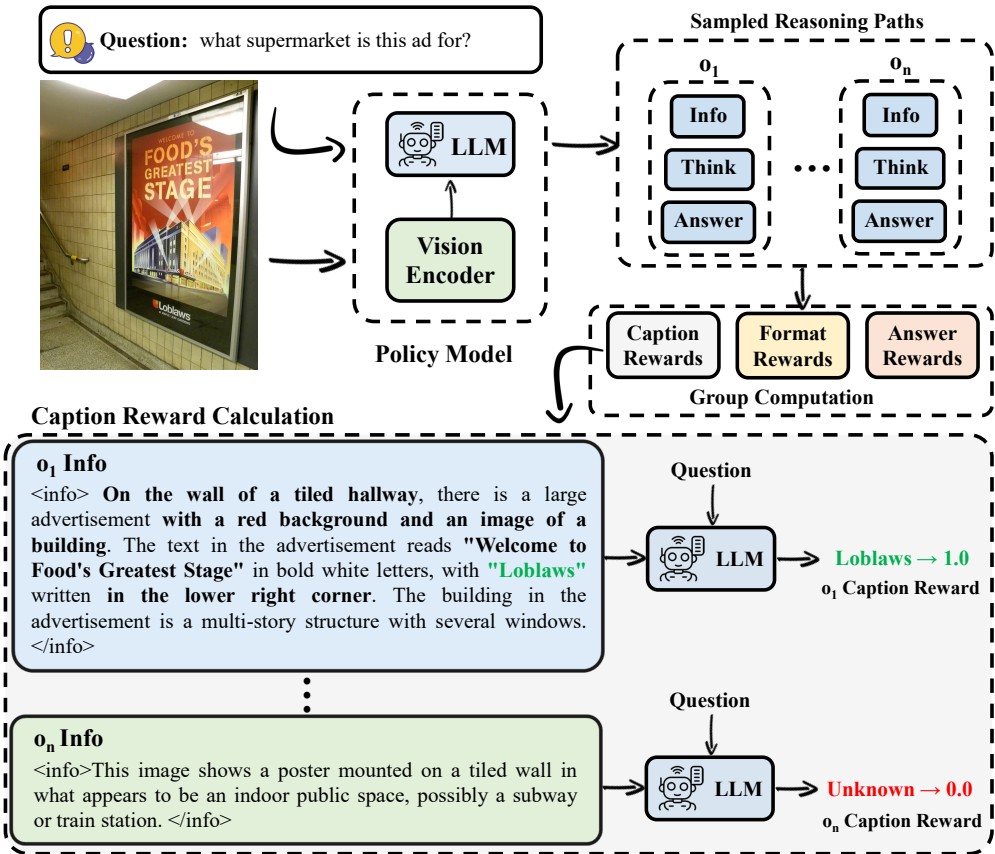

Figure 3: Overview of Visionary-R1. GRPO-style training samples multiple rollouts for each question-answer pair. The format reward checks compliance with the *<info>*–*<think>*–*<answer>* structure, while the caption reward uses the content inside the *<info>* tags only. The caption reward is computed by the policy VLM's own LLM component, not by an external evaluator. All rewards are aggregated to compute the final group-wise advantage.

together, these results support our central claim that robust visual reasoning requires training signals that encourage image-grounded intermediate representations.

### 3.3 Visionary-R1: Improving Reasoning via Captioning

**Caption-Reason-Answer Output Format**    To reduce shortcut exploitation, we require the policy to follow a caption-reason-answer generation process. The underlying principle is to separate image interpretation from downstream inference, so that the reasoning trace operates on an explicit image-derived representation rather than question-side heuristics alone. As illustrated in Fig 3, the output consists of three components:

1. **Caption**: an image-faithful description, enclosed in *<info>* tags, covering objects, text, numbers, attributes, and spatial relations that may be relevant to the question;

2. **Reasoning**: a chain-of-thought trace, enclosed in *<think>* tags, that performs inference based on the image-derived information in the caption;

3. **Answer**: a concise final prediction, enclosed in *<answer>* tags.

Compared with the conventional reason-answer format, this design introduces an inductive bias that makes explicit image interpretation an intermediate step. Yet structural constraints alone are insufficient: a model

> **Policy Model Prompt**
>
> You are tasked with analyzing an image to generate an exhaustive and detailed description. Your goal is to extract and describe all possible information from the image, including but not limited to objects, numbers, text, and the relationships between these elements. The description should be as fine and detailed as possible, capturing every nuance. After generating the detailed description, you need to analyze it and provide step-by-step detailed reasoning for the given question based on the information. Finally, provide a single word or phrase answer to the question. The description, reasoning process and answer are enclosed within <info> </info>, <think> </think> and <answer> </answer> tags, respectively, i.e., <info> image description here </info> <think> reasoning process here </think> <answer> answer here </answer>.

Figure 4: System prompt used for the policy model. The prompt explicitly requires the model to generate an *<info>* description before *<think>* and *<answer>*.

may follow the required format while still producing generic or weakly relevant captions. Visionary-R1 therefore combines the caption-first generation protocol with an explicit caption reward that encourages the information field to be both faithful to the image and useful for answering the question.

During training, a dedicated system prompt requires the policy to generate the *<info>* field before the *<think>* and *<answer>* fields. This prompt-level specification reduces ambiguity in format evaluation and aligns the decoding order with the intended optimization target. As we later show in Section 4.1, prompting alone is not sufficient without reward signals tied to image evidence.

### 3.4 Reward Design

We optimize the policy with three complementary rewards. For a training sample $x = (v, q, a)$, Visionary-R1 generates a rollout $o_i = (r_i, y_i)$, where the reasoning trajectory is instantiated as $r_i = (c_i, t_i)$ with caption $c_i$ and reasoning trace $t_i$. For notational simplicity, we suppress the dependence on the current sample $x$ in the reward terms. The reward components are defined as

$$
\begin{aligned}
r_a(o_i) &= \text{Eval}(y_i, a), \\
r_f(o_i) &= \mathbb{I}\left[o_i \in \mathcal{O}_{\text{fmt}}\right], \\
r_c(o_i) &= \text{Eval}\left(f_{\text{eval}}(q, c_i), a\right),
\end{aligned}
\tag{5}
$$

where $\mathcal{O}_{\text{fmt}}$ denotes the set of outputs that satisfy the required *<info>–<think>–<answer>* format, and $f_{\text{eval}}(q, c_i)$ denotes the answer predicted by a caption-only evaluator that receives the question $q$ and the generated caption $c_i$ but no access to the original image. Accordingly, $r_a$ enforces final-answer correctness, $r_f$ encourages structural compliance, and $r_c$ tests whether the caption alone preserves sufficient evidence to support the answer.

The caption reward is more than a preference for detailed descriptions. A policy can satisfy the output template while still generating vague or weakly image-supported captions. By asking a caption-only evaluator to recover the answer from $c_i$ without access to the original image, $r_c$ acts as an information-sufficiency test: it favors captions that retain answer-relevant visual evidence and discourages superficial descriptions. This design is related to reinforcement learning from AI feedback (Bai et al., 2022), but here the feedback specifically targets image-faithful intermediate representations.

The caption reward and the answer reward also impose complementary constraints. The caption reward encourages the *<info>* field to be informative, whereas the answer reward verifies whether the resulting trajectory supports a correct final prediction. A visually inconsistent but fluent caption is therefore unlikely to consistently maximize the joint objective. To further reduce reward hacking, the caption evaluator is

---

**Caption Reward Prompt**

You are an analytical assistant designed to evaluate texts and answer questions based on strict criteria. Follow these steps:

Analyze the Text: Check if the provided text contains answers, solutions, explanations, problem-solving, or interpretations (e.g., reasoning steps, conclusions, causal statements like "because" or "therefore"). If any such elements exist, classify the text as non-descriptive.

Determine Response: If the text is purely descriptive (e.g., objectively describing images, diagrams, or scenes without explanations/answers), answer the user's question using only the description in a single word or phrase. If the text is non-descriptive, respond with "Hacking Sample".

---

Figure 5: Prompt used to compute the caption reward. The evaluator must answer the question using only the caption content, discouraging hidden reasoning or answer leakage inside the $<info>$ field.

prompted to answer using only the caption content, filtering hidden reasoning or answer leakage inside the information field. This improves alignment between the reward and faithful image interpretation.

The overall reward of rollout $o_i$ is then written as

$$R(o_i) = r_a(o_i) + r_f(o_i) + \alpha\, r_c(o_i), \tag{6}$$

where $\alpha$ controls the contribution of the caption reward. This formulation preserves answer correctness as the primary optimization target while explicitly encouraging image-faithful intermediate representations.

### 3.5 GRPO Training

Group Relative Policy Optimization (GRPO) was introduced in DeepSeekMath (Shao et al., 2024) for text-only reasoning and later adopted in DeepSeek-R1 (Guo et al., 2025a). It simplifies reinforcement learning by eliminating the need for an explicit critic: for each sample, a group of responses is generated and the normalized within-group rewards are used to compute advantages. Relative to standard GRPO, our formulation introduces two modifications for visual reasoning. First, we incorporate the caption reward described above to encourage image-faithful intermediate representations rather than answer-only optimization. Second, we replace a static KL penalty with a cosine-annealed schedule, which stabilizes training while gradually relaxing the constraint to permit richer reasoning trajectories.

**Policy Optimization** For each training sample $x = (v, q, a) \sim \mathcal{D}$, we use the observable input $u = (v, q)$ as the multimodal prompt and sample $n$ rollouts $\{o_1, o_2, \ldots, o_n\}$ from the old policy model $\pi_{\theta_{\text{old}}}(\cdot \mid u)$. Each rollout is scored using the combined reward $R(o_i)$ from Eq 6. Let $\mathbf{R} = \{R(o_1), R(o_2), \ldots, R(o_n)\}$ denote the reward group for the current sample. An advantage value is then computed from this reward group, $\mathbf{R}$, as

$$A_i = \frac{R(o_i) - \text{mean}(\mathbf{R})}{\text{std}(\mathbf{R})}, \quad i = 1, \cdots, n. \tag{7}$$

Let $\rho_i(\theta) = \frac{\pi_\theta(o_i|u)}{\pi_{\theta_{\text{old}}}(o_i|u)}$ denote the importance ratio for the $i$-th rollout. The updated policy $\pi_\theta$ is trained using the clipped surrogate objective

$$\mathcal{J}(\theta) = \mathbb{E}_{x \sim \mathcal{D},\, u=(v,q),\, \{o_i\}_{i=1}^n \sim \pi_{\theta_{\text{old}}}(\cdot|u)} \left[ \frac{1}{n} \sum_{i=1}^n \Big( \min\big(\rho_i(\theta) A_i, \text{clip}\big(\rho_i(\theta), 1 - \varepsilon, 1 + \varepsilon\big) A_i\big) - \beta\, d_{\text{KL}}(o_i, u) \Big) \right], \tag{8}$$

where both $\varepsilon$ and $\beta$ are hyper-parameters. $\varepsilon$ controls the clipping bound and limits the range of policy updates, while $\beta$ is the KL penalty coefficient that regularizes deviation from a reference policy $\pi_{\text{ref}}$ through the per-rollout penalty $d_{\text{KL}}(o_i, u)$ defined below.

Table 2: Composition of the reinforcement learning training corpus.

| Dataset | Size | Answer Type | Visual Format |
|---|---|---|---|
| A-OKVQA (Schwenk et al., 2022) | 17.1K | Multi-choice | General Scene |
| ChartQA (Masry et al., 2022) | 28.3K | Open-text+Num | Chart |
| AI2D (Kembhavi et al., 2016) | 15.5K | Multi-choice | Diagram |
| ScienceQA (Lu et al., 2022a) | 6.2K | Multi-choice | Scene + Chart |
| GeoQA+ (Cao & Xiao, 2022) | 12.1K | Multi-choice | Math |
| DocVQA (Mathew et al., 2021) | 39.5K | Open-text | Document |
| CLEVR-Math (Lindström & Abraham, 2022) | 32.6K | Num | 3D |
| Icon-QA (Lu et al., 2021) | 29.9K | Multi-choice | Diagram |
| TabMWP (Lu et al., 2022b) | 23.1K | Open-text+Num | Table |
| RoBUT SQA (Zhao et al., 2023) | 34.1K | Open-text+Num | Chart |
| TextVQA (Singh et al., 2019) | 34.6K | Multi-choice | General Scene |
| Total | 272.6K | – | – |

**Cosine Annealing KL Coefficient**   The KL penalty regularizes deviation from the reference policy and stabilizes optimization. However, a large static coefficient can overly constrain the policy and suppress detailed reasoning, whereas an overly small coefficient may encourage unstable updates and degenerate reward-maximizing behavior (Skalse et al., 2022). We therefore anneal the KL penalty over time: the coefficient is larger during early training, when optimization is most unstable, and smaller later, when the policy benefits from more freedom to develop richer reasoning traces. The per-rollout KL penalty is written as

$$d_{\mathrm{KL}}(o_i, u) = \frac{\pi_{\mathrm{ref}}(o_i \mid u)}{\pi_\theta(o_i \mid u)} - \log \frac{\pi_{\mathrm{ref}}(o_i \mid u)}{\pi_\theta(o_i \mid u)} - 1. \tag{9}$$

We then replace $\beta$ in Eq 8 with the cosine-annealed coefficient $\hat{\beta}_t$:

$$\hat{\beta}_t = \frac{\beta}{2} \times \left( 1 + \cos\left( \pi \times \frac{t}{T_{\max}} \right) \right), \tag{10}$$

where $t$ and $T_{\max}$ denote the current and maximum training steps, respectively.

## 4 Experiments

### 4.1 Experimental Settings

**Training Data**   Unlike methods that rely on curated reasoning annotations, our approach learns exclusively from CoT-free visual question-answer pairs. To maximize diversity, we aggregate 11 public datasets without dataset-specific filtering or prompt rewriting. The resulting corpus contains 272.6K image-question-answer triplets spanning general scenes, charts, tables, diagrams, mathematical figures, documents, and 3D scenes. As summarized in Table 2, it also covers multi-choice, open-text, and numeric answers. This heterogeneity exposes the policy to diverse sources of visual evidence and reduces the risk that the learned strategy overfits to a single benchmark family.

**Benchmarks**   We evaluate Visionary-R1 on six representative visual reasoning benchmarks: MathVista (Lu et al., 2023), MathVision (Wang et al., 2024), MMBench (Liu et al., 2024), MMMUPro (Yue et al., 2024), MMStar (Chen et al., 2024a), and CV-Bench (Tong et al., 2024). Together, these benchmarks cover algebraic and scientific reasoning, broad multimodal understanding, fine-grained perceptual reasoning, and 2D/3D visual analysis. This protocol measures not only performance on the three primary benchmarks, but also whether the learned reasoning strategy transfers to more diverse and previously unseen tasks.

**Baseline Methods**   To isolate the effect of our design, we implement two in-house baselines. **SFT** performs supervised fine-tuning on the original question-answer pairs and serves as a lower bound without reinforcement learning. **GRPO** applies answer-driven RL using the same backbone and training data, isolating the

Table 3: Comparison with prior methods on three visual reasoning benchmarks. SFT and RL denote supervised fine-tuning and reinforcement learning, respectively. CoT denotes chain-of-thought supervision, either self-generated or distilled from third-party models. QA denotes training with question-answer pairs only. Visionary-R1 uses a 3B model trained with QA-only RL. * indicates results borrowed from the Seed report (Guo et al., 2025b).

|  | Size | Strategy | Data | MathVista | MathVision | MMBench |
|---|---|---|---|---|---|---|
| *Closed-source models* | | | | | | |
| GPT-4o*(Hurst et al., 2024) | - | - | - | 63.8 | 31.2 | 84.3 |
| GPT-o1*(Jaech et al., 2024) | - | - | - | 71.8 | 63.2 | 83.8 |
| Claude3.5-Sonnet (Anthropic, 2024) | - | - | - | 67.7 | 37.9 | 82.6 |
| Claude3.7-Sonnet*(Anthropic, 2025) | - | - | - | 74.5 | 58.6 | 82.0 |
| Gemini-1.5-Pro (Team et al., 2024) | - | - | - | 63.9 | 19.2 | 73.9 |
| Gemini-2.5-Pro*(Google, 2025) | - | - | - | 82.7 | 73.3 | 90.1 |
| *Open-source models* | | | | | | |
| Qwen2.5-VL (Bai et al., 2025) | 3B | - | - | 62.3 | 21.2 | 79.1 |
| InternVL2.5 (Chen et al., 2024b) | 4B | - | - | 60.5 | 20.9 | 81.1 |
| MiniCPM-V2.6 (Yao et al., 2024b) | 8B | - | - | 60.6 | 17.5 | 81.5 |
| LLaMA3.2 (AI, 2024) | 11B | - | - | 51.5 | - | 65.8 |
| *Reasoning models* | | | | | | |
| Ovis (Yan et al., 2025) | 4B | SFT | CoT | 66.6 | - | 79.3 |
| Mulberry (Yao et al., 2024a) | 7B | SFT | CoT | 63.1 | - | - |
| R1-Onevision (Yang et al., 2025) | 7B | SFT+RL | CoT | 64.1 | 29.9 | - |
| Insight-V (Dong et al., 2024) | 7B | SFT+RL | CoT | 59.9 | - | 82.3 |
| R1-VL (Zhang et al., 2025) | 7B | SFT+RL | CoT | 63.5 | 24.7 | - |
| LLaVA-CoT (Xu et al., 2024) | 11B | SFT | CoT | 54.8 | - | 75 |
| *Our models* | | | | | | |
| Base Model | 3B | - | - | 61.5 | 19.1 | 82.1 |
| SFT | 3B | SFT | QA | 54.6 | 7.0 | 80.7 |
| GRPO | 3B | RL | QA | 61.8 | 20.3 | 78.6 |
| Visionary-R1 | 3B | RL | QA | 69.4 | 24.7 | 84.1 |

effect of applying GRPO without our caption-grounding design. We additionally compare with state-of-the-art methods reported in the literature, including proprietary systems (e.g., GPT-4o and Claude3.5) and open-source models (e.g., InternVL2.5 and LLaMA3.2).

**Implementation Details** We adopt Qwen2.5-VL-3B (Bai et al., 2025) as the base model. For group-wise reward computation, we sample $n = 8$ responses per instance with temperature 0.9. All parameters are optimized with a learning rate of $5 \times 10^{-7}$. Unless otherwise specified, the caption-reward weight is set to $\alpha = 0.1$ and the initial KL coefficient is set to $\beta = 0.04$. The prompt images in Figs 4 and 5 are the exact templates used to implement the caption-first policy and the caption-only reward evaluation.

## 4.2 Main Results

The quantitative results are summarized in Tab 3. Supervised fine-tuning on question-answer pairs alone degrades relative to the base model on all three benchmarks, with the largest drop on MathVision. Vanilla GRPO recovers part of this loss and yields modest gains on MathVista and MathVision, but it remains below the base model on MMBench. Together with the qualitative examples in Fig 1 and the grounding analysis in Table 1, these observations indicate that answer-driven RL can improve task performance without reliably inducing grounded visual reasoning.

**Test Data**

**Boxes per shipment**

| Stem | Leaf |
|------|------|
| 1 | 9 |
| 2 | 6 |
| 3 | 4 9 |
| 4 | 1 9 |
| 5 | 0 3 9 |
| 6 | 4 8 |

Q:A shipping company keeps track of the number of boxes in each shipment they send out. How many shipments had exactly 56 boxes? A:0

**GRPO**

<think> To find how many shipments had exactly 56 boxes, we need to look for the number 56 in the plot. </think>
**<answer> 1 </answer>**

**GRPO+Caption**

<info> The image is a stem-and-leaf plot representing the number of boxes in each shipment. The stem represents the tens digit, and the leaf represents the ones digit. </info> <think> To find the number of shipments with exactly 56 boxes, **we need to look at the row where the stem is 5 and the leaf is 6.** So, there is no entry in the row where the stem is 5 and the leaf is 6, which means there is no shipment with exactly 56 boxes. </think> **<answer> 0 </answer>**

**GRPO+Caption+r_c**

<info> The image is a stem-and-leaf plot representing the number of boxes in each shipment. The stem represents the tens digit, and the leaf represents the ones digit. **The plot shows the following data: 19, 26, 34, 39, 41, 49, 50, 53, 59, 64, 68.** </info> <think> To find how many shipments had exactly 56 boxes, we need to look for the number 56 in the plot. **In the stem-and-leaf plot, the number 56 would be represented by the stem 5 and the leaf 6. We can see that 56 is not listed in the plot.** Therefore, there are no shipments with exactly 56 boxes. </think> **<answer> 0 </answer>**

Figure 6: Qualitative comparison of model outputs. Caption-first generation improves response coherence, and the caption reward further increases the visual specificity of both the intermediate caption and the downstream reasoning trace.

Table 4: Additional evaluation on diverse visual reasoning benchmarks.

| Methods | Size | MMMUPro | MMStar | CV-Bench-2D | CV-Bench-3D |
|---------|------|---------|--------|-------------|-------------|
| Base Model | 7B | 42.5 | 48.0 | 69.8 | 54.2 |
| R1-VL | 7B | 29.1 | 60.0 | 67.2 | 65.9 |
| Base Model | 7B | 38.3 | 63.9 | 74.1 | 72.6 |
| R1-Onevision | 7B | 21.9 | 59.1 | 34.2 | 20.1 |
| Base Model | 3B | 31.6 | 52.4 | 72.6 | 71.1 |
| Visionary-R1 | 3B | 34.0 | 61.5 | 74.4 | 74.0 |

Visionary-R1 consistently outperforms the corresponding 3B baselines, improving over the base model by 7.9 points on MathVista, 5.6 points on MathVision, and 2.0 points on MMBench. Despite relying only on question-answer supervision, it is competitive with, and on several benchmarks stronger than, prior reasoning-oriented VLMs and several proprietary systems. Crucially, these gains coincide with a marked reduction in shortcut behavior: Table 1 shows that Visionary-R1 lowers the shortcut score from 69.2% for GRPO to 4.5% while also achieving the highest answer accuracy. This coupling between accuracy and shortcut reduction supports the interpretation that the gains arise from better use of image evidence rather than from superficial format compliance.

To evaluate transfer beyond the three primary benchmarks, we further test on MMMUPro, MMStar, and CV-Bench. As reported in Table 4, Visionary-R1 improves over its 3B base model on all four additional metrics, including both the 2D and 3D splits of CV-Bench. The consistency of these gains suggests that the caption-first objective yields reasoning strategies that transfer across diverse task distributions.

### 4.3 Ablation Study and Analyses

**Effectiveness of Captioning and Caption Reward** We conduct focused ablations to disentangle the contributions of the caption-first design. Starting from GRPO, we incrementally add the caption output format and the caption reward $r_c$. Instead of using the full 272.6K corpus, we train on individual datasets to reduce computation while preserving diversity: 1) training on ChartQA and evaluating on MathVista and MathVision, and 2) training on A-OKVQA and evaluating on MMStar and MMBench.

Table 5: Results under different KL-coefficient schedules. Dynamic schedules improve Visionary-R1 more consistently than static coefficients, with cosine annealing giving the best overall performance.

| Method | Strategy | MathVista | MathVision | MMStar | MMBench |
|---|---|---|---|---|---|
| Visionary-R1 | Static (0.04) | 60.9 | 19.3 | 54.2 | 82.6 |
| | Static (0.008) | 60.7 | 18.7 | 56.0 | 82.7 |
| | Linear | 63.4 | 22.4 | 60.4 | 84.6 |
| | Cosine | 64.6 | 22.7 | 61.6 | 85.5 |
| GRPO | Static (0.04) | 59.0 | 18.2 | 48.1 | 80.4 |
| | Cosine | 59.6 | 18.4 | 46.6 | 80.9 |

Table 6: Ablation of the caption-first design. 'Cap' inserts a caption before reasoning, while 'LR' and 'CR' denote Length Reward and Caption Reward, respectively.

| | Train: ChartQA | | Train: A-OKVQA | |
|---|---|---|---|---|
| Method | MathVista | MathVision | MMStar | MMBench |
| Zero-shot | 61.5 | 19.1 | 52.4 | 82.1 |
| *+Cap* | 60.3 | 18.2 | 51.9 | 79.8 |
| GRPO | 59.0 | 18.2 | 54.2 | 82.6 |
| *+Cap* | 62.6 | 20.9 | 60.4 | 85.5 |
| *+Cap+LR* | 62.0 | 20.3 | 59.6 | 85.2 |
| *+Cap+CR* | **64.6** | **22.7** | **62.9** | **87.6** |

Table 6 shows three clear findings. First, adding a caption field to GRPO already improves generalization, suggesting that separating image interpretation from downstream reasoning is a useful inductive bias. Second, adding the caption reward produces the strongest gains across both settings, confirming that reward signals tied to image evidence are more important than format alone. Third, replacing the caption reward with a simple length reward offers no benefit and can slightly hurt performance, indicating that verbosity by itself does not solve the shortcut problem.

We also compare against a prompt-only zero-shot baseline, where the base model is instructed at inference time to follow the caption-reason-answer format without any RL optimization. As reported in Table 6, this baseline is consistently worse than the original base model, showing that exposure to the output structure alone does not improve reasoning. Together with Table 1, this result strengthens the claim that the gains of Visionary-R1 come from grounded RL optimization rather than from a superficial format prior. Figure 6 provides complementary qualitative evidence: caption-first generation improves coherence, while the caption reward makes both the intermediate caption and the downstream reasoning more visually specific.

**KL Coefficient**   We compare three strategies for choosing the KL coefficient $\beta$: static values, linear decay, and cosine annealing (Eq 10). For the static setting, we evaluate 0.04 and 0.008, where the former follows common practice and the latter tests a weaker regularization regime. As shown in Table 5, static coefficients perform worst, whereas dynamic schedules substantially improve performance; cosine annealing gives the best overall results. Applying cosine annealing to GRPO without the caption reward provides little benefit, suggesting that the schedule is particularly useful when the policy must learn a grounded captioning stage.

To understand this performance gap, we analyze several training dynamics in Fig 7, including output length, format reward, and caption reward. With a static coefficient of 0.04, the output length grows rapidly to an abnormal level around 700 steps, while both the format reward and the caption reward collapse, indicating unstable optimization. The bottom panel of Fig 7 shows that the resulting outputs become long but semantically empty. With a smaller static coefficient of 0.008, the policy instead exploits a degenerate solution (Stiennon et al., 2020): it places a short reasoning trace where the caption should appear and emits

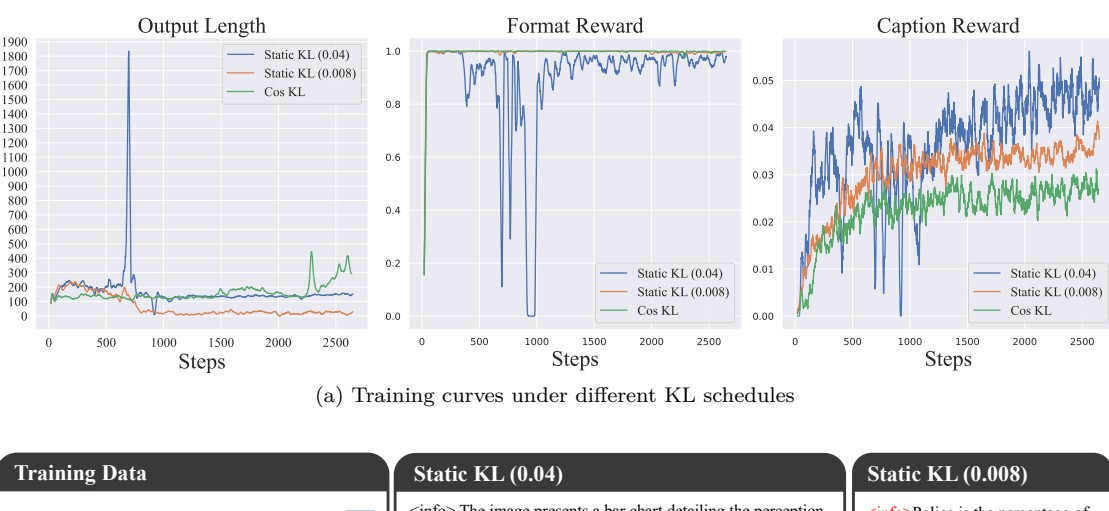

(a) Training curves under different KL schedules

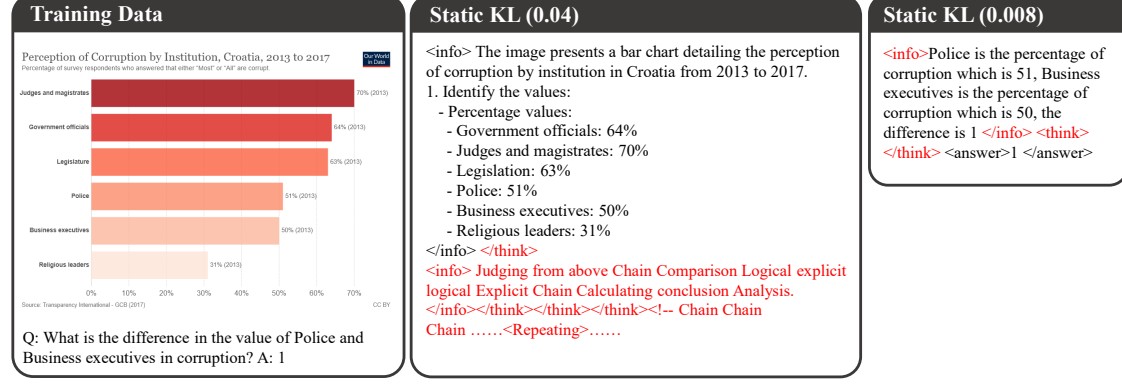

(b) Representative outputs

Figure 7: Training dynamics and representative outputs under different KL-coefficient schedules.

Table 7: Model-scaling results using A-OKVQA (17.1K) for reinforcement learning.

| Methods | Size | RL Data | MathVista | MathVision | MMStar | MMBench |
|---|---|---|---|---|---|---|
| Base Model | 3B | – | 61.5 | 19.1 | 52.4 | 82.1 |
| Visionary-R1 | 3B | 17K | 62.5 | 20.5 | 62.9 | 87.6 |
| Base Model | 7B | – | 68.1 | 22.5 | 63.2 | 83.9 |
| Visionary-R1 | 7B | 17K | 70.2 | 24.4 | 66.7 | 89.5 |

no content inside the *<info>* tags. This behavior increases answer reward without promoting transferable visual reasoning. Both linear decay and cosine annealing alleviate these failure modes by regularizing early training while relaxing the KL constraint later.

**Model Scaling**  We further evaluate whether the proposed training strategy remains effective when the underlying VLM is scaled up. Following the supplementary setting, we train both 3B and 7B models on the A-OKVQA subset containing 17.1K samples. Table 7 shows that Visionary-R1 improves over the base model for both scales on all reported benchmarks. This result suggests that the proposed caption-first RL objective is not tied to a specific model size and can continue to provide gains when stronger visual backbones are used.

**Reward Weight Analysis**  We follow the original recipe by assigning equal weights to the accuracy and format rewards. The caption reward introduces an additional coefficient $\alpha$, raising the question of whether performance is sensitive to its value. Since the caption reward should improve caption quality without overshadowing answer correctness and format consistency, we choose weights smaller than 1 and compare

Table 8: Hyper-parameter analysis for the caption-reward weight $\alpha$.

| Weight $\alpha$ | MathVista | MathVision | MMStar | MMBench |
|:---:|:---:|:---:|:---:|:---:|
| 0.1 | 63.5 | 20.7 | 60.4 | 82.1 |
| 0.5 | 63.1 | 20.8 | 60.4 | 82.5 |

$\alpha = 0.1$ with $\alpha = 0.5$ while keeping the rest of the training recipe unchanged. As reported in Table 8, the two configurations yield highly similar results, suggesting that the method is fairly robust to the precise choice of $\alpha$. We therefore use $\alpha = 0.1$ in all main experiments because it provides stable optimization while preserving the benefit of the caption signal.

**Hallucination Analysis** A natural concern is that introducing intermediate captions and optimizing with a caption reward may encourage the model to fabricate visual evidence. To examine this, we evaluate hallucination and faithfulness on POPE Li et al. (2023) and HumbleBench Tong et al. (2025). POPE tests object hallucination with yes/no questions, while HumbleBench measures epistemic humility over object, relation, and attribute queries, providing direct diagnostics beyond standard accuracy.

Table 9: Hallucination and faithfulness comparison on POPE and HumbleBench.

| Model | # Params | POPE | HumbleBench | | | |
|:---|:---:|:---:|:---:|:---:|:---:|:---:|
| | | Overall ↑ | Object ↑ | Relation ↑ | Attribute ↑ | Overall ↑ |
| LLaVA-CoT Xu et al. (2024) | 11B | - | 60.85 | 65.09 | 73.68 | 66.79 |
| R1-Onevision Yang et al. (2025) | 7B | 83.1 | 61.43 | 65.16 | 73.38 | 66.89 |
| R1-VL Zhang et al. (2025) | 7B | 85.7 | 63.59 | **67.96** | 74.03 | 68.73 |
| **Visionary-R1** | 3B | **87.8** | **65.75** | 66.70 | **75.88** | **69.65** |

As shown in Table 9, Visionary-R1 does not exhibit a high hallucination risk compared with other methods. On POPE, our 3B model achieves 87.8, outperforming R1-VL-7B by 2.1 points and R1-Onevision-7B by 4.7 points. On HumbleBench, Visionary-R1 obtains the best overall score of 69.65, improving over R1-VL-7B and R1-Onevision-7B by 0.92 and 2.76 points, respectively. It also performs best on the Object and Attribute subsets, where hallucinated objects and unsupported attributes are directly penalized.

These results suggest that caption insertion and the caption reward do not lead to more fabricated visual claims. Instead, they act as grounding signals that encourage the model to align intermediate descriptions with visible evidence before reasoning.

**Inference Efficiency Analysis** Besides accuracy, inference efficiency is also important for practical visual reasoning systems. We therefore compare different models on the MathVista benchmark under a fixed inference protocol. All models are evaluated with the same decoding and post-processing settings using the vLLM backend on a single NVIDIA H20 GPU. We set the temperature to 0 to ensure deterministic generation and stable latency measurement. For the Qwen2.5-VL baselines, we use the standard CoT prompt, "Please think step by step and then provide the final answer." We report three metrics: answer accuracy, average number of generated output tokens, and average wall-clock inference time per sample. This controlled setting allows us to compare the accuracy–efficiency trade-off without introducing adaptive routing or model-specific inference strategies.

The results show that Visionary-R1 provides a favorable accuracy–efficiency trade-off. Compared with the same-size Qwen2.5-VL-7B CoT baseline, Visionary-R1-7B improves accuracy by 5.5 points while reducing output tokens by 30.2% and latency by 56.1%. Even Visionary-R1-3B outperforms the Qwen2.5-VL-7B CoT baseline by 4.7 points while being 2.6× faster. Against reasoning-heavy 7B models, Visionary-R1-7B uses substantially fewer tokens and achieves substantially lower latency. For instance, compared with Vision-R1-7B, Visionary-R1-7B reduces output tokens by 29.2% and latency by 56.2%, though Vision-R1-7B obtains higher accuracy. Similar trends hold for VL-Rethinker-7B and Solution-back-7B, which achieve

Table 10: Efficiency comparison under the same setting. Qwen baselines use the CoT prompt "Please think step by step and then provide the final answer."

| Model | Acc. (%) ↑ | Avg. output tokens ↓ | Time/sample (s) ↓ |
|---|---|---|---|
| Qwen2.5-VL-3B w/ CoT Prompt | 55.6 | 174.5 | 0.695 |
| Qwen2.5-VL-7B w/ CoT Prompt | 64.7 | 228.0 | 1.376 |
| Vision-R1-7B Huang et al. (2025) | 73.5 | 224.7 | 1.380 |
| VL-Rethinker-7B Wang et al. (2026) | 74.9 | 271.1 | 1.552 |
| Solution-back-7B Yang et al. (2026) | 72.3 | 279.1 | 1.642 |
| Visionary-R1-3B | 69.4 | 137.4 | 0.526 |
| Visionary-R1-7B | 70.2 | 159.2 | 0.604 |

strong performance but require longer generations and higher latency. These results indicate that Visionary-R1 occupies a different point on the accuracy–efficiency frontier: it targets visually grounded reasoning at a much lower inference cost.

## 5 Conclusion

This paper identifies shortcut learning as a central obstacle when applying reinforcement learning to visual language models. Our results show that answer-level rewards alone are insufficient to induce transferable visual reasoning: a policy can achieve competitive accuracy while remaining weakly grounded in the image. Visionary-R1 addresses this issue by rewarding caption-first intermediate representations, enabling QA-only reinforcement learning to produce more grounded and generalizable reasoning behavior.

Experiments demonstrate consistent gains across multiple benchmarks and model scales, while the cosine-annealed KL schedule further stabilizes multimodal RL training. One limitation of the current framework is its reliance on an auxiliary caption evaluator, whose quality may influence reward fidelity. Exploring alternative grounded intermediate objectives and extending the approach to broader multimodal settings are promising directions for future work.

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

# A   Technical Appendices and Supplementary Material

## A.1   Evaluator Consistency for Shortcut Measurement

To assess whether the proposed shortcut score is robust to the choice of evaluator, we conduct an additional consistency analysis using multiple independent VLM judges. Specifically, we re-evaluate the same set of sampled rollouts with four evaluators, GPT-5, Claude-4.5-Sonnet, Gemini-3-Flash, and Qwen2.5-VL-72B, while keeping the evaluation rubric, input format, and scoring protocol fixed across all judges. Each evaluator is asked to estimate the image–reasoning relevance score, from which the shortcut score is computed as its complement. Thus, all evaluators measure the same quantity under the same criterion, and only the judge model is varied.

Table 11 reports the resulting shortcut scores. Although the absolute scores differ across evaluators, the relative ordering remains highly consistent. In all four cases, the GRPO-failure subset receives the highest shortcut score, indicating that failed predictions are most strongly associated with image-agnostic or weakly grounded reasoning. The GRPO-all and GRPO-correct subsets obtain lower but still substantial shortcut scores, with GRPO-correct consistently below GRPO-all. In contrast, Visionary-R1 obtains much lower shortcut scores under every evaluator. Its scores are approximately an order of magnitude smaller than those of the GRPO variants, suggesting that its reasoning traces are substantially more grounded in visual evidence.

These results show that our main conclusion does not depend on a single evaluator. The consistency of the ranking across four independent judges supports the reliability and reproducibility of the shortcut indicator, despite moderate differences in score calibration among evaluators.

## A.2   Prompt Template for Shortcut Evaluation

To further facilitate reproducibility, we provide the exact prompt template used for shortcut evaluation. All evaluator models use the same prompt, rubric, and output format. Given an image, the corresponding question, and the model-generated reasoning trace, the evaluator is instructed to assess only the degree to which the reasoning is grounded in visible image evidence. The evaluator is explicitly asked not to judge the final answer correctness, since the purpose of this diagnostic is to measure image–reasoning relevance rather than task accuracy. The shortcut score is then computed as the complement of the returned relevance score, i.e., $1 - s_{\mathrm{vis}}$.

Table 11: Consistency of shortcut scores ($\downarrow$) across independent evaluators. We re-evaluate the same rollouts with four VLM judges under an identical rubric and report the mean and standard deviation across evaluators.

| Evaluator | GRPO-all | GRPO-correct | GRPO-failure | Visionary-R1 |
|---|---|---|---|---|
| GPT-5 | 69.2 | 65.1 | 84.7 | 4.5 |
| Claude-4.5-sonnet | 64.9 | 61.0 | 79.5 | 7.2 |
| Gemini-3-flash | 73.7 | 69.2 | 90.5 | 2.8 |
| Qwen2.5-VL-72B | 67.1 | 63.0 | 82.4 | 6.3 |
| Mean $\pm$ Std. | $68.7 \pm 3.7$ | $64.6 \pm 3.5$ | $84.3 \pm 4.7$ | $5.2 \pm 2.0$ |

---

**Prompt Template for Shortcut Evaluation**

**System Instruction.**
You are an impartial evaluator. Your task is to assess whether the given reasoning trace is grounded in the visual evidence of the image. You should evaluate image–reasoning relevance only. Do not evaluate whether the final answer is correct.

**Input.**
You are given:

- an image;
- a question about the image;
- a reasoning trace generated by a vision-language model.

**Evaluation Goal.**
Please determine how strongly the reasoning trace is supported by visible evidence in the image. A reasoning trace should receive a high score if its key claims and intermediate steps can be verified from the image. It should receive a low score if it mainly relies on textual cues from the question, generic commonsense priors, answer-frequency biases, or unsupported assumptions that could be made without looking at the image.

**Scoring Rubric.**
Assign an image–reasoning relevance score $s_{\text{vis}} \in [0, 1]$ according to the following rubric:

- 1.0: The reasoning is strongly grounded in the image. The key visual claims are directly supported by visible evidence.
- 0.75: The reasoning is mostly grounded in the image, with only minor unsupported or generic statements.
- 0.5: The reasoning uses some image evidence, but also relies substantially on assumptions, question-side cues, or generic priors.
- 0.25: The reasoning is weakly related to the image and is mostly driven by textual cues or superficial patterns.
- 0.0: The reasoning is unrelated to the image, ignores the visual content, or makes unsupported visual claims.

**Important Notes.**

- Evaluate visual grounding, not final answer correctness.
- Penalize reasoning that could plausibly be generated without seeing the image.
- Penalize hallucinated objects, attributes, spatial relations, numbers, chart values, or other visual details not supported by the image.
- If the reasoning contains both grounded and ungrounded steps, assign an intermediate score.
- Use the full range from 0 to 1 when appropriate.

**Output Format.**
Return the result in the following JSON format only:

```
{"score":  <a number between 0 and 1>, "rationale":  "<brief explanation>"}
```

**Question:**
{question}

**Generated Reasoning Trace:**
{reasoning}

---

Figure 8: Prompt template used for shortcut evaluation. All evaluators receive the same image, question, generated reasoning trace, scoring rubric, and JSON output instruction.

### A.3 Visualization of the Visionary-R1 Output

To illustrate the validity of Visionary-R1 across various visual formats, we visualized the output for each format individually. The results are presented in Fig. 9-15.

**Document**

Q: Where the seminar is conducted?
A: Snow Lake Lodge, Mount Snow, Vermont

**Visionary-R1**

<info> The image shows a black and white document that appears to be a invitation or informational flyer. The details of the seminar are as follows:

- Title: Third Day Seminar
- Topic: Epidemiology & Prevention of Cardiovascular Diseases
- Organizer: Council on Epidemiology of the American Heart Association
- Collaborator: The National Heart, Lung & Blood Institute
- Date: July 30 - August 12, 1977
- Location: Snow Lake Lodge, Mount Snow, Vermont

The flyer has a minimalistic design with a logo featuring a heart and a torch at the top-left corner, which could be associated with the American Heart Association, given their logo often resembles a flame or torch within a heart shape. </info> <think> Given the information on the document, the key aspect to focus on is the location provided, which is explicitly mentioned as "Snow Lake Lodge, Mount Snow, Vermont." </think> <answer> Snow Lake Lodge, Mount Snow, Vermont </answer>

Figure 9: Visualization of Visionary-R1 Output in Document Format.

**General Scene**

**Visionary-R1**

<info> In the image, there is a narrow dirt road in a rural setting, surrounded by lush greenery and tropical plants. A person is riding a motorcycle down the road, while another person is walking beside the motorcycle. There are also a cow is visible walking alongside the motorcycle. The street is appears to be in a village or countryside area. </info>

<think> The street in the image is dirt, indicating that it has not been paved. Paved streets are typically made from materials such as asphalt, concrete, or gravel, which are more resistant to weathering and wear. The lack of a paved surface suggests that this is a rural area where construction and maintenance of paved roads may not be as frequent or feasible. </think> <answer> no one </answer>

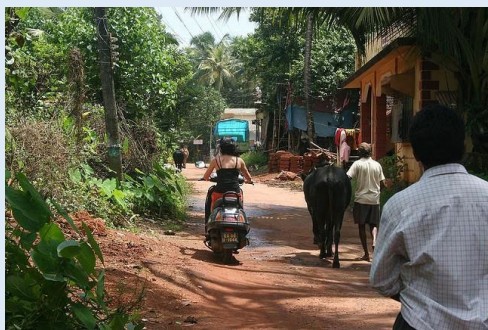

Q: Who paved this street?
A: no one

Figure 10: Visualization of Visionary-R1 Output in General Scene Format.

**Table**

| tilapia | $4 per kilogram |
|---------|-----------------|
| catfish | $4 per kilogram |
| salmon  | $2 per kilogram |
| halibut | $3 per kilogram |
| tuna    | $4 per kilogram |
| trout   | $3 per kilogram |

Q: Manny wants to buy 3+9/10 kilograms of salmon. How much will he spend?
A: 7.8

**Visionary-R1**

<info> The image provides a list of fish prices per kilogram:
- Tilapia: $4 per kilogram
- Catfish: $4 per kilogram
- Salmon: $2 per kilogram
- Halibut: $3 per kilogram
- Tuna: $4 per kilogram
- Trout: $3 per kilogram
</info>
<think>
To calculate the total cost, we multiply the price per kilogram of salmon ($2) by the number of kilograms Manny wants to buy (3.9 kilograms).
The calculation is as follows:
Price per kilogram * Number of kilograms = Total cost
$2 * 3.9 = $7.80 </think>
<answer>
$7.80
</answer>

Figure 11: Visualization of Visionary-R1 Output in Table Format.

**3D**

**Visionary-R1**

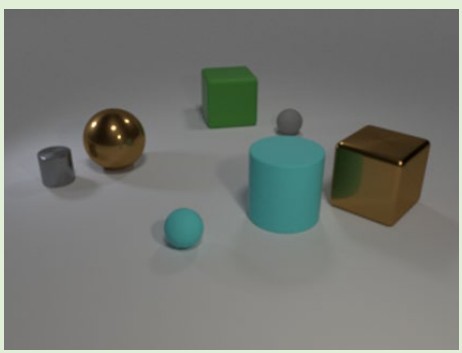

Q: Subtract all large green matte cubes. Subtract all big green blocks. How many objects are left?
A: 6

<info> The image contains a variety of geometric shapes and objects. There is a large brown metallic sphere, a small gray metallic cylinder, a small gray matte sphere, a large green matte cube, a small cyan matte sphere, a large cyan matte cylinder, and a large brown metallic cube. </info>
<think> To determine how many objects are left after subtracting the large green matte cube and the big green block, we need to count the remaining objects. The large green matte cube is the same as the big green block, so we subtract one of each. This leaves us with the brown metallic sphere, the gray metallic cylinder, the small gray matte sphere, the cyan matte sphere, the cyan matte cylinder, and the brown metallic cube. Counting these, we find there are 6 objects left. </think>
<answer> 6 </answer>

Figure 12: Visualization of Visionary-R1 Output in 3D Format.

**Chart**

**Visionary-R1**

Q: What is the predicted increase in online sales of electricals and electronics by 2018?
A: 23.2

<info>
The image is a bar chart with two data series: Offline sales and Online sales. The x-axis represents the years 2013 and 2018, while the y-axis represents the share of retail sales as a percentage.
For the year 2013, the Online sales are shown at 17.5%, and Offline sales are 82.5%.
For 2018, the Online sales increased to 23.2%, and Offline sales decreased to 76.8%.
It's worth noting that the source of the data is Statista 2021.
</info>
<think> The increase in online sales is shown in the percentage, from 17.5 percent in 2013 to 23.2 percent in 2018.
</think>
<answer> 23.2 </answer>

Figure 13: Visualization of Visionary-R1 Output in Chart Format.

**Math**

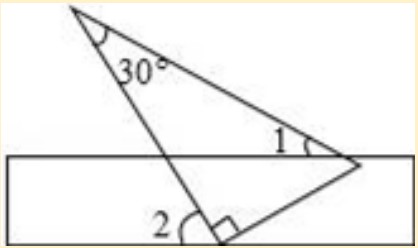

Q: Place a ruler and a right-angled triangle ruler as shown in the figure (∠ACB is a right Angle). Given that ∠1=15°, what is the size of ∠2?
A: 45°

**Visionary-R1**

<info> A ruler and a right-angled triangle board. The right-angled triangle board has a 30° Angle. There are two marks on the ruler. One is in the middle and the other is in the lower left corner. The Angle indicated by the mark at the lower left corner is ∠2, which is calculated by the Angle between the right Angle of the right-angled triangle and the ruler. The Angle marked in the middle is ∠1. </info>
<think> According to the problem statement, one Angle of a right triangle is 30 degrees and the other Angle is 90 degrees. The problem states that ∠1=15 degrees. According to the formula for the sum of interior angles of a triangle, the other Angle (the Angle adjacent to the right triangle) is 60 degrees. Because the triangle set is a right triangle, the adjacent angles are 30 degrees. A right triangle has a right Angle of 90 degrees. So the Angle that ∠2 is adjacent to 90 degrees of a right triangle is 60 degrees minus 15 degrees, that is, 45 degrees. </think> <answer> 45° </answer>

Figure 14: Visualization of Visionary-R1 Output in Math Format. * *The original input and output were both in Chinese, and we have translated them directly without any modifications.*

**Diagram**

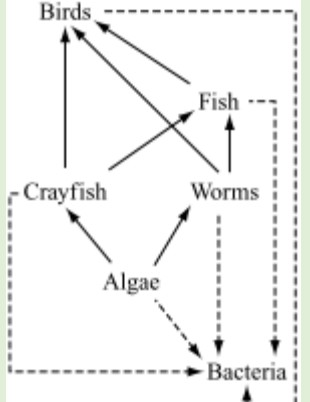

Q: Which is the top predator in the food web? Options: Birds, Crayfish, Fish, None of the above
A: Birds

**Visionary-R1**

<info> This image depicts a food web where organisms are connected by arrows indicating the direction of energy flow from one organism to another. The food web includes various organisms such as birds, fish, crayfish, worms, algae, and bacteria. The arrows show the feeding relationships among these organisms, with the arrows pointing from the prey to the predator. The food web forms a hierarchical structure where energy flows from the bottom to the top, starting from the bacteria and algae at the base and moving through various levels of consumers to the top predators. </info>
<think> To determine the top of the food chain, we need to identify the organisms that do not have any arrows pointing towards them. In this food web, birds are positioned at the top, with arrows pointing towards them from the organisms below. There are no arrows pointing towards birds from the bottom of the web. </think>
<answer> Birds </answer>

Figure 15: Visualization of Visionary-R1 Output in Diagram Format.

