# OpenReview forum: "Visionary-R1: Mitigating Shortcuts in Visual Reasoning with Reinforcement Learning"
_TMLR — Decision pending for TMLR_

### Review · Reviewer_oLcJ · 2026-04-20

**Summary Of Contributions:**

The paper introduces Visionary R1, a framework designed to train visual language models using reinforcement learning and visual question answer pairs without relying on explicit step by step supervision. The authors discovered that applying standard reward optimization often leads models to develop reasoning shortcuts, resulting in poor generalization on complex tasks. To address this issue, they implemented a structured caption reason answer format. This approach forces the model to generate a detailed image description before reasoning, guided by an AI generated caption reward. The resulting model demonstrates superior performance on various visual reasoning benchmarks compared to several commercial models.

Strength:
1. The method propose a strategy of “understanding the image context before reasoning,” which forces the model to comprehend the image context before performing reasoning.
2. The paper is well-formatted and easy to follow.
3. The experiments show the finetuned 3B Qwen2.5VL model outperforms baseline methods.

Weakness:
1. The biggest concern is the literature and baseline models are outdated. The paper lacking of key comparisons with existing works Vision-R1[1] etc. [2][3], which all use Qwen2.5VL as backbone.
2. The reliance on external AI evaluator to compute the caption reward introduces significant risks of biases, reward leakage, and self confirmation bias, as the model might exploit this setup to achieve high rewards without explicitly preventing visual hallucinations.
3. What is the inferece cost compared to other methods? Including captions in the output reasoning chain will bring computational overehad.

[1] Vision-R1: Incentivizing Reasoning Capability in Multimodal Large Language Models, ICLR 26
[2] Look-Back: Implicit Visual Re-focusing in MLLM Reasoning, AAAI 26
[3] VL-Rethinker: Incentivizing Self-Reflection of Vision-Language Models with Reinforcement Learning NeurIPS25

**Audience:**

Yes

**Audience Explanation:**

The topic is popular.

**Broader Impact Concerns:**

NA.

**Claims And Evidence:**

Yes

**Claims Explanation:**

The paper covers details of the experiments and extensive experiments are conducted.

**Requested Changes:**

See weakness.

---

> ### Author Response · Authors · 2026-06-01
> **Response to Reviewer oLcJ**
>
> Thank you for recognizing that the paper is well-formatted, easy to follow, and the performance gains are sizable. We address the three concerns below.
>
> ## 1. Recent related work and baseline comparison
>
> Thank you for pointing out Vision-R1, Look-Back, and VL-Rethinker. We agree that these recent RL-based VLM reasoning works should be discussed and compared. We have added the corresponding discussion and comparison to the Related Work section in the revised manuscript. In addition, Table 10 reports the performance and efficiency comparison with these methods, further clarifying their different assumptions and contributions.
>
> ## 2. Caption evaluator bias and hallucination risk
>
> We agree that an LLM-based evaluator may introduce hallucination risk. To address this concern, we have added hallucination-specific benchmarks. As shown in Table 9, Visionary-R1 obtains \(87.8\) on POPE, outperforming R1-VL-7B by \(2.1\) points and R1-Onevision-7B by \(4.7\) points. On HumbleBench, Visionary-R1 reaches the best overall score (\(69.65\)), with clear gains on object and attribute questions.
>
> These results suggest that the caption reward does not simply increase hallucinated or stylistically persuasive outputs. We further support this with the ablation in Table 6: replacing the caption reward with a length reward is consistently worse, showing that the gain is not due to longer responses alone.
>
> **Table 9. Hallucination comparison on POPE and HumbleBench.**
> *Newly added in the revised manuscript.*
>
> | Model | Params | POPE Overall ↑ | HumbleBench Object ↑ | HumbleBench Relation ↑ | HumbleBench Attribute ↑ | HumbleBench Overall ↑ |
> |:--|:--:|--:|--:|--:|--:|--:|
> | LLaVA-CoT | 11B | - | 60.85 | 65.09 | 73.68 | 66.79 |
> | R1-Onevision | 7B | 83.1 | 61.43 | 65.16 | 73.38 | 66.89 |
> | R1-VL | 7B | 85.7 | 63.59 | **67.96** | 74.03 | 68.73 |
> | **Visionary-R1** | **3B** | **87.8** | **65.75** | 66.70 | **75.88** | **69.65** |
>
> **Table 6. Ablation of the caption-first design on four benchmarks.**
> Adding the caption format alone (Cap) helps over GRPO, but the gain mainly comes from the caption reward (CR); replacing CR with a length reward (LR) does not match CR.
>
> | Method | MathVista (%) ↑ | MathVision (%) ↑ | MMStar (%) ↑ | MMBench (%) ↑ |
> |:--|--:|--:|--:|--:|
> | GRPO | 59.0 | 18.2 | 54.2 | 82.6 |
> | GRPO+Cap | 62.6 | 20.9 | 60.4 | 85.5 |
> | GRPO+Cap+LR | 62.0 | 20.3 | 59.6 | 85.2 |
> | **GRPO+Cap+CR** | **64.6** | **22.7** | **62.9** | **87.6** |
>
> ## 3. Inference cost
>
> Table 10 complements the accuracy--efficiency analysis. Under the same evaluation setting, Visionary-R1-7B improves over the Qwen2.5-VL-7B CoT baseline by \(5.5\) accuracy points while reducing output tokens by \(30.2\%\) and latency by \(56.1\%\). Compared with Vision-R1-7B, Visionary-R1-7B uses \(29.2\%\) fewer tokens and \(56.2\%\) lower latency; Vision-R1 represents a higher-accuracy, higher-cost alternative in this comparison.
>
>
> **Table 10. Efficiency comparison under the same setting.**
> *Newly added in the revised manuscript.*
> Qwen baselines use the CoT prompt "Please think step by step and then provide the final answer."
>
> | Model | Acc. (%) ↑ | Avg. output tokens ↓ | Time/sample (s) ↓ |
> |:--|--:|--:|--:|
> | Qwen2.5-VL-3B w/ CoT Prompt | 55.6 | 174.5 | 0.695 |
> | Qwen2.5-VL-7B w/ CoT Prompt | 64.7 | 228.0 | 1.376 |
> | Vision-R1-7B | 73.5 | 224.7 | 1.380 |
> | VL-Rethinker-7B | 74.9 | 271.1 | 1.552 |
> | Solution-back-7B | 72.3 | 279.1 | 1.642 |
> | **Visionary-R1-3B** | **69.4** | **137.4** | **0.526** |
> | **Visionary-R1-7B** | **70.2** | **159.2** | **0.604** |

---

### Review · Reviewer_4hJJ · 2026-05-18

**Summary Of Contributions:**

This paper studies reinforcement learning for visual reasoning without chain-of-thought supervision and identifies a “shortcut” failure mode when applying vanilla GRPO to VLMs: models tend to emit short, uninformative reasoning that suffices on easy training questions but fails to generalize. The authors propose Visionary-R1, which enforces a caption-reason-answer generation protocol and introduces a caption-only reward computed by an evaluator that must answer based solely on the generated caption, plus a cosine-annealed KL schedule. Trained on ~273K QA-only samples with a 3B backbone, Visionary-R1 shows consistent gains over SFT and GRPO on MathVista, MathVision, MMBench, and additional benchmarks, and the authors present a diagnostic “shortcut score” suggesting markedly improved image-grounding.

**Strengths**

1. The paper studies an important failure mode of RL-based VLM reasoning: models may learn shortcuts instead of relying on visual evidence.

2. The caption–reason–answer design is simple and intuitive, and it provides a clear way to encourage image-grounded reasoning.

3. The experiments are fairly broad, with comparisons to SFT/GRPO baselines and ablations on caption reward and KL scheduling.


**Weaknesses**

1. **The central claim about shortcut learning is not fully convincing.**
   The paper attributes the failure of vanilla GRPO mainly to shortcut learning, but the evidence does not fully rule out other explanations such as unstable RL optimization, insufficient exploration, reward design issues, or the composition of the training data. Although the paper introduces a shortcut score, this metric relies on an external VLM evaluator, so its reliability and correlation with true visual grounding remain unclear. A stronger causal analysis is needed to justify that shortcut learning is indeed the primary source of the observed failures.

2. **The novelty of the proposed caption-before-reasoning framework appears limited.**
   The core idea of asking a model to describe an image before answering is intuitive, but closely related to prior work on caption-assisted VQA and intermediate visual descriptions. The paper should better distinguish Visionary-R1 from these existing approaches and clarify whether the main contribution is the output format, the caption reward, the RL training recipe, or their combination.

3. **The caption reward may introduce reward leakage or self-confirmation bias.**
   The method depends on an auxiliary evaluator to judge whether the generated caption contains enough information to answer the question. However, this reward can be imperfect: captions may include implicit answers, hallucinated evidence, or stylistic cues that satisfy the evaluator without being faithfully grounded in the image. The paper does not sufficiently analyze how often the caption reward misfires or whether the model exploits this reward during training.

4. **Improved accuracy does not necessarily prove improved visual grounding.**
   The paper argues that Visionary-R1 mitigates shortcuts and produces more grounded reasoning, but most evidence still comes from benchmark accuracy and qualitative examples. There is limited evaluation on dedicated grounding, hallucination, or faithfulness benchmarks. Without human evaluation, visual attention analysis, or hallucination-specific tests, it remains unclear whether the model truly reasons from image evidence or simply produces longer and better-formatted responses.

5. **The efficiency cost of always generating captions and long reasoning chains is underexplored.**
   Visionary-R1 forces the model to generate detailed captions before reasoning, even for simple questions. This may substantially increase inference latency and token cost. Since some reported gains may come from longer outputs, the paper should analyze the accuracy–efficiency trade-off and consider whether captioning should be triggered conditionally only for harder examples.

6. **The ablation and robustness studies are not fully sufficient.**
   Although the paper includes several ablations, some important factors are not disentangled enough, such as the separate effects of the format reward, caption reward weight, KL schedule, output length, and dataset composition. The experiments also lack variance across multiple runs, making it difficult to assess the robustness of the reported gains. More comprehensive ablations and statistical analysis would strengthen the empirical claims.

**Audience:**

Yes

**Audience Explanation:**

This topic on improving RL methods for MLLMs is important for the research community.

**Claims And Evidence:**

Yes

**Claims Explanation:**

The method and claim are supported by their main and ablation experiments, mainly compared with GRPO.

**Requested Changes:**

Check the weaknesses in **Summary Of Contributions**. Clarifications and extra results are recommended.

---

> ### Author Response · Authors · 2026-06-01
> **Response to Reviewer 4hJJ (1/3)**
>
> Thank you for the careful feedback and for recognizing that our work studies "*an important failure mode of RL-based VLM reasoning*", uses a "*simple and intuitive*" caption--reason--answer design, and provides "*fairly broad*" experiments.
>
> ## 1. Shortcut learning: scoped claim and reliability checks
>
> You are right that vanilla GRPO can be affected by RL instability, exploration, reward design, and data composition. Our argument is not that shortcut learning is the only cause of failure, but that it is a concrete and measurable VLM-specific failure mode in answer-driven RL. We do not rule out other factors; instead, we show that this failure mode is substantial enough to be isolated and mitigated, which is the motivation of Visionary-R1.
>
> To make the shortcut evaluation more reliable, we re-evaluate the same rollouts with four independent judges using the same rubric. Although absolute scores vary across judges, the ordering by shortcut severity is stable: GRPO-failure exhibits the most shortcut behavior, followed by GRPO-all and GRPO-correct, while the shortcut scores of Visionary-R1 are an order of magnitude lower across all four judges. This supports the reliability of our shortcut indicator and shows that it is not an artifact of any single evaluator.
>
> **Table 11. Shortcut scores (↓) under four independent evaluators.**
> *Newly added in the revised manuscript.*
>
> | Evaluator | GRPO-all ↓ | GRPO-correct ↓ | GRPO-failure ↓ | Visionary-R1 ↓ |
> |:--|--:|--:|--:|--:|
> | GPT-5 | 69.2 | 65.1 | 84.7 | 4.5 |
> | Claude-4.5-sonnet | 64.9 | 61.0 | 79.5 | 7.2 |
> | Gemini-3-flash | 73.7 | 69.2 | 90.5 | 2.8 |
> | Qwen2.5-VL-72B | 67.1 | 63.0 | 82.4 | 6.3 |
> | **Mean ± Std.** | **68.7 ± 3.7** | **64.6 ± 3.5** | **84.3 ± 4.7** | **5.2 ± 2.0** |
>
> ## 2. Novelty: not caption-before-answer alone
>
> The novelty of Visionary-R1 is not the output format alone. Instead, we built an RL-from-AI-feedback (RLAIF) framework to train the caption-reason-answer model without using CoT data, achieving consistent reasoning improvements on multiple challenging benchmarks.
>
> The contribution is the combination of: (i) diagnosing shortcut learning in answer-driven VLM RL; (ii) using a caption reward (CR) to test whether the generated caption preserves enough visual evidence to answer the question, while the answer reward checks the final prediction against ground truth; and (iii) combining this reward with GRPO under a cosine KL schedule, which we find necessary to avoid degenerate solutions---either over-long traces or short, weakly grounded ones. The ablation below shows that format alone is insufficient, while CR provides the main gain.

---

> ### Author Response · Authors · 2026-06-01
> **Response to Reviewer 4hJJ (2/3)**
>
> ## 3. Caption reward leakage and self-confirmation bias
>
> This is a fair concern: a caption-only evaluator can in principle be exploited by answer leakage, hallucinated evidence, or stylistic cues. Concretely, three design choices limit reward hacking: (1) the evaluator sees only the question and the generated caption, not the image or the model's answer; (2) the caption reward is weighted much lower than the answer reward, so the overall optimization remains primarily driven by reasoning and final-answer accuracy, with the caption serving only as an auxiliary grounding signal; and (3) the reward explicitly prohibits including reasoning steps or the final answer in the caption field; if the answer is directly mentioned during caption generation, the caption reward is set to zero.
>
> These design choices are supported by two empirical checks. First, CR consistently outperforms the length-reward variant on all four benchmarks (e.g., MathVista \(62.0 \rightarrow 64.6\), MMStar \(59.6 \rightarrow 62.9\)), showing that the reward is not merely selecting longer or more polished outputs. Second, the hallucination-focused results in Table 9 show that the trained model does not gain accuracy by becoming less faithful: Visionary-R1 improves POPE and HumbleBench over stronger 7B reasoning baselines.
>
> **Table 6. Ablation of the caption-first design on four benchmarks.**
> Adding the caption format alone (Cap) helps over GRPO, but the gain mainly comes from the caption reward (CR); replacing CR with a length reward (LR) does not match CR.
>
> | Method | MathVista (%) ↑ | MathVision (%) ↑ | MMStar (%) ↑ | MMBench (%) ↑ |
> |:--|--:|--:|--:|--:|
> | GRPO | 59.0 | 18.2 | 54.2 | 82.6 |
> | GRPO+Cap | 62.6 | 20.9 | 60.4 | 85.5 |
> | GRPO+Cap+LR | 62.0 | 20.3 | 59.6 | 85.2 |
> | **GRPO+Cap+CR** | **64.6** | **22.7** | **62.9** | **87.6** |
>
> ## 4. Grounding evidence beyond benchmark accuracy
>
> Thanks for your suggestion. We have therefore added dedicated hallucination diagnostics to examine whether the caption reward might be exploited through fabricated visual evidence. Specifically, POPE evaluates object hallucination with yes/no questions, while HumbleBench assesses whether the model avoids over-confident claims across object, relation, and attribute questions. The results help verify that the caption reward does not push the model to invent non-existent visual content simply to obtain a higher reward.
>
> **Table 9. Hallucination comparison on POPE and HumbleBench.**
> *Newly added in the revised manuscript.*
>
> | Model | Params | POPE Overall ↑ | HumbleBench Object ↑ | HumbleBench Relation ↑ | HumbleBench Attribute ↑ | HumbleBench Overall ↑ |
> |:--|:--:|--:|--:|--:|--:|--:|
> | LLaVA-CoT | 11B | - | 60.85 | 65.09 | 73.68 | 66.79 |
> | R1-Onevision | 7B | 83.1 | 61.43 | 65.16 | 73.38 | 66.89 |
> | R1-VL | 7B | 85.7 | 63.59 | **67.96** | 74.03 | 68.73 |
> | **Visionary-R1** | **3B** | **87.8** | **65.75** | 66.70 | **75.88** | **69.65** |
>
> On POPE, our 3B model obtains \(87.8\), outperforming R1-VL-7B by \(2.1\) points and R1-Onevision-7B by \(4.7\) points. On HumbleBench, Visionary-R1 achieves the best overall score (\(69.65\)), improving over R1-VL-7B by \(0.92\) and R1-Onevision-7B by \(2.76\). These results help address the concern that the caption reward may inflate accuracy through hallucinated or unfaithful captions: under explicit hallucination probes, Visionary-R1 shows stronger faithfulness.

---

> ### Author Response · Authors · 2026-06-01
> **Response to Reviewer 4hJJ (3/3)**
>
> ## 5. Accuracy--efficiency trade-off
>
> Thank you for raising this important point. We agree that always producing additional visual descriptions or long reasoning traces may increase token cost and latency, and that conditional captioning is a promising direction. To quantify the accuracy--efficiency trade-off, we have added an efficiency comparison on MathVista. All models are evaluated under the same protocol using vLLM on a single NVIDIA H20 GPU, with identical decoding and post-processing settings. We set temperature to \(0\) for deterministic generation. For Qwen2.5-VL baselines, we use the CoT prompt "Please think step by step and then provide the final answer." We report accuracy, average output tokens, and wall-clock time per sample.
>
> **Table 10. Efficiency comparison under the same setting.**
> *Newly added in the revised manuscript.*
> Qwen baselines use the CoT prompt "Please think step by step and then provide the final answer."
>
> | Model | Acc. (%) ↑ | Avg. output tokens ↓ | Time/sample (s) ↓ |
> |:--|--:|--:|--:|
> | Qwen2.5-VL-3B w/ CoT Prompt | 55.6 | 174.5 | 0.695 |
> | Qwen2.5-VL-7B w/ CoT Prompt | 64.7 | 228.0 | 1.376 |
> | Vision-R1-7B | 73.5 | 224.7 | 1.380 |
> | VL-Rethinker-7B | 74.9 | 271.1 | 1.552 |
> | Solution-back-7B | 72.3 | 279.1 | 1.642 |
> | **Visionary-R1-3B** | **69.4** | **137.4** | **0.526** |
> | **Visionary-R1-7B** | **70.2** | **159.2** | **0.604** |
>
> The results show that our gains are not simply caused by longer outputs. Compared with the same-size Qwen2.5-VL-7B CoT baseline, Visionary-R1-7B improves accuracy from \(64.7\%\) to \(70.2\%\), while reducing output tokens from \(228.0\) to \(159.2\) and latency from \(1.376\,s\) to \(0.604\,s\), corresponding to \(30.2\%\) fewer tokens and \(56.1\%\) lower latency. Even Visionary-R1-3B outperforms the 7B CoT baseline by \(4.7\) points and is \(2.6\times\) faster. Compared with reasoning-heavy models, Visionary-R1 also has much lower inference cost: for example, relative to Vision-R1-7B, Visionary-R1-7B uses \(29.2\%\) fewer tokens and \(56.2\%\) lower latency, although Vision-R1-7B achieves higher accuracy. Similar trends hold for VL-Rethinker-7B and Solution-back-7B, which achieve strong accuracy but require longer generations and higher latency.
>
> These results indicate that our method offers a different accuracy--efficiency trade-off: it aims to obtain visually grounded reasoning with substantially lower inference cost, rather than relying primarily on longer reasoning chains. We agree that adaptive System-1/System-2 routing or conditional captioning could further improve this trade-off, but it is orthogonal to the main focus of this work. We have added this efficiency analysis to the revised paper and discussed conditional captioning as future work.
>
> ## 6. Ablation coverage and robustness
>
> The current experiments are designed to disentangle several relevant factors, including output format and caption reward (Table 6), length reward versus caption reward, KL schedule (static vs. linear vs. cosine), reward weight (\(\alpha=0.1\) vs. \(0.5\)), dataset subsets (ChartQA and A-OKVQA), and model scale (3B and 7B). While these studies do not exhaust all possible sources of variance, they help isolate the contribution of the caption structure and caption reward. We acknowledge that repeated RL training runs are computationally expensive; therefore, following common practice in RL experiments, we fix all random seeds across settings to ensure that the comparisons remain meaningful and controlled.

---

### Review · Reviewer_kNZA · 2026-05-25

**Summary Of Contributions:**

This paper provides a formal identification and mitigation of "shortcut learning" in VLMs optimized via answer-driven RL (particularly, GRPO). To prevent models from exploiting the textual heuristics, the author proposed Visionary-R1, a training framework that structurally enforces the model to do captioning, reasoning, and answering pipeline generation to separate visual perception from logical inference. It is then optimized during RL using a proposed caption reward, which utilizes an independent evaluator to verify that the intermediate caption explicitly captures sufficient visual evidence to solve the problem. The author also introduces a cosine-annealed KL penalty schedule to stabilized the GRPO optimization to prevent reward-hacking. The empirical results show that the training method achieves competitive performance on small scale VLMs against larger proprietary models on multiple visual reasoning benchmark datasets.

Pros:

- The empirical results demonstrate competitive results against large proprietary models on various benchmarks with small scale VLMs
- The ablation study demonstrates some interesting observation, such as the contribution of the captioning task introduction, and the impact of the KL schedule.

Cons:

- The core idea of letting the model describe the image first is a known prompting technique. There are already many papers using such techniques in VQA, such as [R1, R2], the authors didn't discuss. While the proposed method applies it to the VLM training with reinforcement learning, the novelty itself is incremental, as the methodology is theoretically an adaptation of GRPO with a typical captioning bottleneck.
- The proposed method heavily relies on a big proprietary model (GPT-5 here) as the external evaluator, which will introduce massive training latency and cost. The computational overhead and inference cost from the evaluator introduced by the proposed methods were not discussed or quantified against other methods. Moreover, the paper lacks of an ablation on training the model using a cheaper, weaker evaluator as the reward model, which makes the approach's generalizability questionable.

Refs:

- [R1] Kapuriya, Janak, et al. "Enhancing Scientific Visual Question Answering via Vision-Caption aware Supervised Fine-Tuning." Proceedings of the 2nd International Workshop on Large Vision-Language Model Learning and Applications. 2025.
- [R2] Özdemir, Övgü, and Erdem Akagündüz. "Enhancing visual question answering through question-driven image captions as prompts." Proceedings of the IEEE/CVF Conference on Computer Vision and Pattern Recognition. 2024.

**Audience:**

No

**Audience Explanation:**

Given the weak novelty compared to existing framework and the usage of typical GRPO framework, and the heavy reliance on the expensive external model as the evaluator, the existing findings might not be as exciting to the audience.

**Claims And Evidence:**

No

**Claims Explanation:**

1. The authors claim the method "aims to lower the development cost of training VLMs for visual reasoning" by eliminating the distillation of CoT supervision. However, the proposed method uses a huge external proprietary model (GPT-5) to evaluate the caption reward which introduces massive computational overhead and cost. The paper didn't quantify these cost and compare it against baselines, making the claim of cost-saving unsupported.
2. The caption-reasoning-answer format proposed in this work is not a novel structure as the technique of enforcing the VLM describing the image before reasoning or answering the questions are well-documented in VQA and prompting literature.
3. Without the ablation study on using different models as the evaluator, the paper didn't prove that the methodology is robust or generalizable.

**Requested Changes:**

1. The author should articulate the key difference and contribution of the proposed method against those prompting and fine-tuning techniques that enforce the model to describe before answering, explaining why it is not a simple A + B where the existing technique is simply applied in a RL framework.
2. The training cost, and latency overhead of using such a big model during the training stage should be quantified and compared to the CoT training baselines
3. An ablation of the reward model, especially with a smaller, localized, or open-sourced model, should be conducted to demonstrate the generalization ability or gap of the proposed method.

---

> ### Author Response · Authors · 2026-06-01
> **Response to Reviewer kNZA**
>
> Thank you for the constructive comments. We first clarify a key misunderstanding: **GPT-5 is not used as the caption-reward evaluator in our RL training**. It is used only for the offline shortcut-score analysis reported in the paper. Therefore, GPT-5 is not in the training loop, does not score captions during RL, and does not contribute to training latency or proprietary-model cost.
>
> ## 1. Novelty beyond caption-before-answer prompting
>
> Our contribution goes beyond simply asking the model to describe the image before answering. Instead, we first identify shortcut learning as a measurable failure mode of answer-driven VLM RL: the policy can obtain reward while systematically under-using visual evidence. We then quantify this behavior and summarize its causes, which naturally motivates captioning as an interpretable way to make visual evidence explicit before prediction. In our RL-from-AI-feedback (RLAIF) framework, this caption step is part of the training procedure rather than a standalone prompting trick: we further design a caption reward to encourage the intermediate description to preserve answer-relevant visual evidence, without relying on CoT supervision or external models. The ablations in Table 6 support this design: caption formatting alone helps but is insufficient, and a simple length reward cannot replace the proposed caption reward. **Together, this observation--analysis--solution pipeline shows that our method is a complete and interpretable system, rather than a simple A+B combination.**
>
> **Table 6. Ablation of the caption-first design on four benchmarks.**
> Adding the caption format alone (Cap) helps over GRPO, but the gain mainly comes from the caption reward (CR); replacing CR with a length reward (LR) does not match CR.
>
> | Method | MathVista (%) ↑ | MathVision (%) ↑ | MMStar (%) ↑ | MMBench (%) ↑ |
> |:--|--:|--:|--:|--:|
> | GRPO | 59.0 | 18.2 | 54.2 | 82.6 |
> | GRPO+Cap | 62.6 | 20.9 | 60.4 | 85.5 |
> | GRPO+Cap+LR | 62.0 | 20.3 | 59.6 | 85.2 |
> | **GRPO+Cap+CR** | **64.6** | **22.7** | **62.9** | **87.6** |
>
>
> ## 2. Training and inference cost
>
> **We emphasize that our RL training does not query GPT-5 or any other external large model.** The caption reward is computed with the lightweight LLM component associated with the VLM itself, so the reward computation is self-contained rather than dependent on a proprietary evaluator. For inference, no reward evaluator is used at all. As shown in Table 10, under the same evaluation setting, Visionary-R1 uses fewer output tokens and lower latency than CoT-style baselines while maintaining competitive accuracy.
>
> **Table 10. Efficiency comparison under the same setting.**
> *Newly added in the revised manuscript.*
> Qwen baselines use the CoT prompt "Please think step by step and then provide the final answer."
>
> | Model | Acc. (%) ↑ | Avg. output tokens ↓ | Time/sample (s) ↓ |
> |:--|--:|--:|--:|
> | Qwen2.5-VL-3B w/ CoT Prompt | 55.6 | 174.5 | 0.695 |
> | Qwen2.5-VL-7B w/ CoT Prompt | 64.7 | 228.0 | 1.376 |
> | Vision-R1-7B | 73.5 | 224.7 | 1.380 |
> | VL-Rethinker-7B | 74.9 | 271.1 | 1.552 |
> | Solution-back-7B | 72.3 | 279.1 | 1.642 |
> | **Visionary-R1-3B** | **69.4** | **137.4** | **0.526** |
> | **Visionary-R1-7B** | **70.2** | **159.2** | **0.604** |

---

### Author Response · Authors · 2026-06-01
**Summary of Revisions**

## Main Concerns and Responses

- **Caption reward.** We clarify that the caption reward does not rely on GPT-5 or any external proprietary model during RL training. It is computed by the lightweight LLM component associated with the VLM itself, making the reward computation self-contained. We further address reward hacking and hallucination concerns through ablations and hallucination diagnostics: the caption reward outperforms a length reward, and Visionary-R1 improves POPE and HumbleBench, suggesting that the reward does not simply encourage longer, fabricated, or stylistically persuasive captions.
- **Efficiency.** We add controlled comparisons with CoT-prompted baselines and recent RL-based VLM reasoning methods. These results show that Visionary-R1 achieves a favorable accuracy--efficiency trade-off, obtaining competitive performance with substantially fewer output tokens and lower latency than reasoning-heavy baselines.
- **Novelty.** We emphasize that the contribution is not merely a caption-before-answer format. Instead, our work forms a complete observation--analysis--solution pipeline: we identify shortcut learning as a new measurable failure mode in answer-driven VLM RL, quantify and analyze it, and mitigate it through caption-first RLAIF with a caption reward. This provides a new perspective for RL training of VLMs by explicitly encouraging grounded intermediate representations.

## Overall Changes in the Revised Manuscript (marked blue in the new revisions)

- Expanded the Related Work with Vision-R1, VL-Rethinker, and Look-Back, clarifying their use of distilled reasoning traces, trigger-style deliberation, or external CoT data, and contrasting them with our QA-only shortcut-mitigation setting.
- Clarified the shortcut-measurement discussion by adding multi-evaluator reliability checks and by emphasizing that GPT-5 captions are used only for the offline diagnostic intervention, not for RL training.
- Added evaluator-consistency results with four VLM judges---GPT-5, Claude-4.5-Sonnet, Gemini-3-Flash, and Qwen2.5-VL-72B---showing stable shortcut-score rankings across evaluators.
- Added the exact shortcut-evaluation prompt in the appendix, including the image--reasoning relevance rubric, JSON output format, and instruction to evaluate grounding rather than answer correctness.
- Added a hallucination analysis section and Table 9 on POPE and HumbleBench, showing that Visionary-R1 does not increase fabricated visual evidence and improves over prior reasoning baselines on faithfulness diagnostics.
- Added an inference-efficiency analysis section and Table 10, comparing CoT-prompted Qwen baselines, Vision-R1, VL-Rethinker, Solution-back, and Visionary-R1 under the same MathVista/vLLM/H20 protocol.
- Reported the revised efficiency findings: Visionary-R1 achieves a favorable accuracy--efficiency trade-off, reducing output tokens and latency substantially relative to CoT and reasoning-heavy 7B baselines while maintaining competitive accuracy.

---

### Decision · Action_Editor_5dpb · 2026-07-10

**Recommendation:** Accept as is

**Audience:**

Yes

**Audience Explanation:**

The paper tackles the timely problem of reinforcement learning for visual reasoning, an active research area with growing interest in the machine learning and computer vision communities.

**Claims And Evidence:**

Yes

**Claims Explanation:**

This paper proposes a reinforcement learning framework (referred to Visionary-R1) for training visual language models (VLMs) to perform visual reasoning using only image–question–answer pairs, without explicit chain-of-thought (CoT) supervision. To mitigate the shortcut learning problem observed when directly applying GRPO to VLMs, the authors introduce a caption–reason–answer generation paradigm, which encourages the model to first generate a detailed image caption before performing reasoning. Trained with reinforcement learning on 273K CoT-free visual question-answer pairs, Visionary-R1 achieves competitive performance across multiple visual reasoning benchmarks.

Reviewers appreciated the comprehensive experimental evaluation and the empirical performance of the proposed approach. In response to the reviews, the authors further strengthened the evaluation by adding analyses of reward hacking and hallucination, along with controlled comparisons against CoT-prompted baselines and recent RL-based VLM reasoning methods.

In their original reviews, the reviewers raised several common concerns, primarily regarding the overall technical novelty relative to existing baselines, the reliance on an external AI evaluator, and the increased inference latency and token cost introduced by the  process. The rebuttal adequately addressed these concerns and provided additional empirical evidence supporting the effectiveness and robustness of the proposed framework.